# Identification of Six Novel Proteins Containing a ZP Module from Nemertean Species

**DOI:** 10.3390/biom14121545

**Published:** 2024-12-02

**Authors:** Jumpei Ikenaga, Kaoru Yoshida, Manabu Yoshida

**Affiliations:** 1Misaki Marine Biological Station, School of Science, The University of Tokyo, Koajiro 1024, Misaki, Miura 238-0225, Kanagawa, Japan; 2Faculty of Biomedical Engineering, Toin University of Yokohama, Yokohama 225-8503, Kanagawa, Japan; yoshidak@toin.ac.jp

**Keywords:** Nemertea, *Kulikovia*, fertilization, ZP, EGF-like, evolution

## Abstract

During fertilization, a series of reactions between the eggs and spermatozoa proceed predominantly in a species-specific manner. The molecules mediating these species-specific reactions remain unknown except in a few organisms. In this study, we focused on two species belonging to the phylum Nemertea, *Kulikovia alborostrata* and *K. fulva*, and explored molecules involved in species-specific interactions between gametes. Orthologs of molecules known to be involved in species-specific reactions were not expressed in the ovaries of these two species. In contrast, we identified six novel proteins, named NeZPL1–NeZPL6, containing a ZP module. Among these, we found that NeZPL6 is located on the surface of an unfertilized egg and is suggested to be involved in its interaction with spermatozoa. Furthermore, we found an indel of three amino acids in the EGF-like domain of NeZPL6, which possibly confers species specificity to this interaction. Our results suggested the existence of a novel system for species recognition in animal gametes.

## 1. Introduction

Fertilization consists of a series of processes, including sperm chemotaxis, acrosome reaction, penetration into the zona pellucida (ZP) in mammals or the vitelline envelope (VE) in other species, and sperm–egg fusion. In most cases, these processes occur in a species-specific manner and are mediated by species-specific interactions between molecules on the gametes [1]. ZP module-containing proteins are particularly known to play crucial roles in gamete interactions. The ZP module consists of a ZP-C domain, a ZP-N domain, and a linker region connecting these two domains [2]. ZP module-containing proteins polymerize through their ZP-C and ZP-N domains to form filaments, which constitute the structural basis of ZP or VE [3,4,5,6]. These proteins usually contain a consensus furin cleavage site (CFCS, -R-X-K/R-R-) and are processed by furin during incorporation into the ZP structure [7,8]. The N-terminal region of ZP module-containing proteins protrudes from the polymerized filament and varies among proteins, conferring unique functions to each protein [9]. For example, ZP2 (a component of the mammalian ZP) and VERL (a component of the abalone VE) are ZP domain-containing proteins that play a role in species-specific interactions between gametes [10,11]. The ZP2 contains three ZP-N domains in its N-terminal region, which are believed to be essential for species-specific interactions with unidentified molecules in sperm [10]. VERL contains 22 repeated ZP-N-like domain sequences prior to the ZP module [12]. Although the sequences of the ZP-N-like domains in VERL and the ZP-N domains of ZP2 are not very similar, they share a common structural fold [11]. Among the 22 repeats in VERL, the first and second are under positive selection, and, particularly, the second is involved in species-specific interactions with lysin on sperm [11].

The animals of the phylum Nemertea, commonly known as “ribbon worms”, belong to the superphylum Lophotrochozoa sensu stricto, along with the phyla Phoronida and Brachiopoda, based on phylogenetic analysis of whole-genome sequences [13]. Except for some terrestrial and freshwater nemertean species, all lophotrochozoan species inhabit marine environments. Most marine nemerteans undergo external fertilization; therefore, species-specific interactions between gametes are crucial for the maintenance of these species. However, to date, no studies have focused on the molecular mechanisms underlying species-specific interactions during nemertean fertilization. Two nemertean species, *Kulikovia alborostrata*, and *Kulikovia fulva*, are suitable models for investigating this phenomenon because their gametes possess a species-recognition mechanism that prevents crossbreeding despite their genetic proximity [14]. Because the morphology of gametes of these species is similar [14], the mechanism can be attributed to a molecular difference on the surface of gametes.

In the present study, we investigated the proteins involved in species recognition in the gametes of *K. alborostrata* and *K. fulva*. We identified six novel ZP domain-containing proteins (NeZPL1–NeZPL6), some of which were expressed in maturing oocytes. A comparison of the sequences of the orthologs of the two species revealed the presence of the three-aminoacid indel in NeZPL6.

## 2. Materials and Methods

### 2.1. Specimens Collection, Gametes Collection, and Fertilization

*Kulikovia alborostrata* and *K. fulva* were collected from calcareous algae in Moroiso and Arai-hama in Kanagawa, Japan. Sampling was conducted during the spring tide in February 2020 and 2021. One male and one female were used for the cross-fertilization experiment. To collect gametes, the posterior end of the specimen was cut (approximately 3 mm in length) with a razor blade. The cut fragments were then transferred to glass dishes containing filtered seawater (FSW). The spermatozoa that leaked from the cut fragments of the male were collected using a pipette and transferred to a microtube. The oocytes released from the cut fragments of the female were transferred to another glass dish containing FSW. The oocytes were incubated for about an hour at 18 °C to induce maturation. The sperm concentrations were calculated using a hemocytometer. Fertilization was conducted in a plastic dish containing 2 mL FSW and approximately 50 mature eggs. The fertilization rate was calculated 2 h after insemination by counting the eggs during cleavage.

### 2.2. RNA Extraction and cDNA Synthesis

Total RNA was extracted from an oocytes-containing fragment of *K. alborostrata* and *K. fulva* using an RNeasy Micro Kit (QIAGEN, Aarhus, Denmark). The extracted RNA was used for the transcriptome analysis. Single-strand cDNA was synthesized with SMARTer^®^ RACE 5′/3′ Kit (TaKaRa, Kusatsu, Japan).

### 2.3. cDNA Representational Difference Analysis

We extracted total RNA from fragments containing ovaries of both mature females and immature females (the same individual but after spawning) and synthesized single-stranded cDNA as above. Double-strand cDNA was prepared from the single-strand cDNA with the 5′ PCR primer II A in SMART™ PCR cDNA Synthesis Kit (TaKaRa, Kusatsu, Japan) The second strand was synthesized by polymerase chain reaction (PCR) using ExTaq (Takara, Kusatsu, Japan). The PCR products were purified using the phenol/chloroform extraction method, followed by ethanol precipitation. The purified DNA was dissolved in 10 µL of distilled water (DW), followed by the addition of 10 µL of 10× H buffer, 1 µL of Sau3AI (Takara, Kusatsu, Japan), and 79 µL of DW. This mixture was incubated at 37 °C for 7 h and then purified. The following procedure conformed to that described in a previous study [15]. The second-difference products were cloned into the pGEM-T (Easy) plasmid (Promega, Madison, WI, USA), and the successfully inserted plasmids were selected by colony-direct PCR with M13F and M13R primers. The PCR products were purified with Exonuclease I (New England Biolabs, Ipswich, MA, USA) and rAPid alkaline phosphatase (Roche, Mannheim, Germany) and then used for sequencing reaction with BigDye™ Terminator v3.1 (Thermo Fisher Scientific, Waltham, MA, USA). To obtain the full length of the products, we performed both 5′ and 3′ RACE using SMARTer^®^ RACE 5′/3′ Kit (TaKaRa, Kusatsu, Japan) with the single-strand cDNA and primers designed using the partial sequences obtained by the cDNA RDA. Predicted genes coded by those products were identified by BLAST search with the database of *Notospermus geniculatus* [13].

### 2.4. Searching Genes Coding a ZP-Module in Transcripts

We performed de novo assembly of RNA-seq reads from the mature ovary with Trinity v2.14.0 [16]. Then, we searched genes coding a ZP-module using Sequenceserver v2.0.0 [17] with an amino acid sequence of the ZP-module in the obtained product by cDNA RDA as a query. The identified genes were also confirmed with RACE with cDNA from mature ovary using SMARTer^®^ RACE 5′/3′ Kit (TaKaRa, Kusatsu, Japan).

### 2.5. In Situ Hybridization

To synthesize the probes, partial gene sequences were amplified by PCR using the primers listed in Appendix A and GoTaq (Promega, Madison, WI, USA). The products were purified using NucleoSpin Gel and PCR Clean-up (TaKaRa, Kusatsu, Japan) or QIAquick Gel Extraction Kit (QIAGEN, Aarhus, Denmark). Purified PCR products were cloned into the plasmid pGEM T (Easy) (Promega, Madison, WI, USA) and transformed into Stellar competent cells (TaKaRa, Kusatsu, Japan). Plasmids were purified using a QIAquick Plasmid QuickPure Kit (QIAGEN, Aarhus, Denmark). Templates for the probe were prepared by PCR using the M13F and M13R primers. The products were purified using NucleoSpin Gel and PCR Clean-up (TaKaRa, Kusatsu, Japan). The probes were synthesized with T7 or SP6 polymerase (Roche, Mannheim, Germany) at 37 °C for 15 min and purified using ethanol precipitation with LiCl2. The method for whole-mount in situ hybridization was originally described for the ascidian *Ciona intestinalis* (*C. robusta*) embryos by Ikuta et al. [18] and was modified to fit *K. alborostrata* eggs in the present study. Small pieces were excised from the posterior end of a ripe female of *K. alborostrata* and fixed in 4% PFA, 0.5 M NaCl, and 0.1 M MOPS (pH 7.2) for 24 h at 4 °C. The samples were then washed twice with PBST. After washing, proteinase K treatment was performed by 2 µg/µL proteinase K (QIAGEN, Aarhus, Denmark) at 37 °C for 15 min, followed by the post-fixation in 4% PFA for 20 min and washing with PBST twice. The samples were pre-incubated in pre-hybridization buffer (4×SSC, 50% formamide, 1× Denhardt’s, 100 µg yeast RNA, and 0.1% Tween 20) for 10 min and then incubated in fresh pre-hybridization buffer at 60 °C for 2 h. After the pre-hybridization, the samples were transferred into 100 µL of pre-hybridization buffer containing 50 ng of probe and incubated at 60 °C for 16 h. Then, the samples were washed twice with 2×SSC, 50% formamide, and 0.1% Tween 20 at 60 °C for 15 min, followed by washing with solution A (0.5 M NaCl, 10 mM Tris-HCl pH8.0, 5 mM EDTA, and 0.1% Tween 20) thrice. Next, the samples were incubated in solution A containing 20 µg/µL RNase A at 37 °C for 30 min and washed in solution A at once. After washing with 2×SSC, 50% formamide, and 0.1% Tween 20 at 60 °C for 20 min, with 0.5×SSC, 50% formamide, 0.1% Tween 20 at 60 °C for 15 min twice and with PBST twice, blocking was carried out with 1% BlockAce (KAC, Kyoto, Japan) at room temperature for 30 min. The samples were soaked in 1% BlockAce containing anti-DIG AP (1/2000 dilution) (Roche, Mannheim, Germany) at room temperature for 2 h. Before the final coloring reaction in TMNT containing 4 µL NBT/BCIP (Roche, Mannheim, Germany), the samples were washed in PBST once and in TMNT twice.

### 2.6. In Vitro mRNA Synthesis and Microinjection

NeZPL6 from *K. alborostrata* and *K. fulva* were amplified by PCR using the primers listed in Appendix A. The PCR products were purified with NucleoSpin Gel and PCR Clean-up (TaKaRa, Kusatsu, Japan) and cloned into pRACE plasmid in SMARTer^®^ RACE 5′/3′ Kit (TaKaRa, Kusatsu, Japan) with In-Fusion^®^ HD Cloning Kit (TaKaRa, Kusatsu, Japan). The plasmids were transformed into Stellar competent cells (TaKaRa, Kusatsu, Japan), and successfully inserted plasmids were selected by colony-direct PCR. The selected plasmids were purified using a QIAquick Plasmid QuickPure Kit (QIAGEN, Aarhus, Denmark). To insert EGFP behind the signal peptide in NeZPL6, a linearized plasmid was prepared by PCR using primers (KaZP6F × KaKfZP6R or KfZP6F × KaKfZP6R) and KOD One (TOYOBO, Osaka, Japan). The products were treated with DpnI (Takara, KusatsuShiga, Japan) at 37 °C for 4 h and purified by QIAEXII (QIAGEN, Aarhus, Denmark). EGFP was amplified with primers (KaGFPZP6F × KaGFPZP6R or KfGFPZP6F × KfGFPZP6R) from pEGFP-C1 plasmid. The product was purified using NucleoSpin Gel and PCR Clean-up (TaKaRa, Kusatsu, Japan). Then, the linearized plasmid and EGFP were ligated with In-Fusion^®^ HD Cloning Kit (TaKaRa, Kusatsu, Japan), and plasmids were collected as described above. To prepare the template for mRNA synthesis, PCR was conducted using primers M13R and ZP6R. mRNA was synthesized using the mMessage mMachine SP6 Kit (Thermo Fisher Scientific, Waltham, MA, USA) and the poly(A) Tailing Kit (Thermo Fisher Scientific, Waltham, MA, USA), followed by phenol/chloroform extraction and isopropanol precipitation. The purified mRNA was dissolved in DW (100 ng/µL) and microinjected into oocytes, which had been kept immature by transferring them into Ca^2+^-free ASW just after spawning [19]. The oocytes were derived from a single female. After microinjection, oocytes were transferred to FSW and incubated for 2 h to induce maturation through germinal vesicle breakdown (GVBD) before observing fluorescence with confocal microscopy FV3000 (Olympus, Tokyo, Japan). The plasma membranes of the eggs were stained with PlasMem Bright Red (Dojindo, Kumamoto, Japan).

## 3. Results

### 3.1. Cross-Fertilization Ability Between K. alborostrata and K. fulva

We evaluated the cross-fertilization ability between the gametes of *K. alborostrata* and *K. fulva* under various sperm concentrations (Figure 1). Almost all eggs inseminated with conspecific sperm were fertilized, regardless of the sperm concentration. In contrast, the fertilization rate of eggs inseminated by heterospecific sperm did not exceed 24% even at a sperm concentration of 1.0 × 10^4^ cells/mL and decreased as sperm concentration decreased. These results were consistent with the previous study [14]. Considering that the sperm concentration in the natural environment rarely reaches 1.0 × 10^2^ cells/mL due to the immediate diffusion of sperm, this species-recognition mechanism is likely sufficient to maintain reproductive isolation between the two species.

### 3.2. Six Novel ZP Module-Containing Proteins in K. alborostrata and K. fulva

To identify the genes involved in species recognition, we first performed a cDNA representational difference analysis (cDNA RDA) using cDNA from the mature and immature ovary of *K. alborostrata* for the purpose of extracting genes highly expressed in the ovary of *K. alborostrata*. As shown in Table 1, the gene encoding the ZP module was identified in transcripts from the mature ovary as the product of cDNA RDA. Next, we performed BLAST analysis on de novo assembled transcripts from the ovaries of *K. alborostrata* and *K. fulva* to seek other genes coding the ZP module because it is known that ZP module-containing proteins are incorporated with other ZP module-containing proteins in ZP of vertebrates [6]. The sequence of the ZP module obtained by cDNA RDA was used as a query. We identified six ZP module-coding genes in both species, which we named NeZPL1–NeZPL6 after Nemertean ZP-like module containing proteins (Figure 2). We confirmed these genes actually existed in cDNA from mature ovaries by RACE. All of these genes were predicted to contain a signal peptide and a transmembrane region. NeZPL1, NeZPL4, NeZPL5, and NeZPL6 have one or two EGF-like domains in addition to a ZP module. NeZPL2 has eight repetitive CUB domains, and NeZPL3 has a VWD domain in addition to a ZP module. All the NeZPL genes contained a CFCS immediately after the ZP module. In contrast, other ZP proteins with structures homologous to ZP2 or VERL were not found in the transcripts of the two species.

### 3.3. Comparison Between Orthologs of NeZPL Genes

A comparison of the NeZPL1–NeZPL6 sequences between *K. alborostrata* and *K. fulva* revealed several nonsynonymous substitutions (Appendix A). Notably, a three-amino acid indel was found in the second EGF-like domain of NeZPL6. *K. fulva* possesses proline, aspartic acid, and methionine sequences that are absent in *K. alborostrata* (Figure 3A and Appendix A). This indel occurs in the loop formed by the first and third cysteines of the typical EGF-like domain (Figure 3B).

### 3.4. Expression of NeZPL1–NeZPL6 in Oocyte

We examined the expression of NeZPL1–NeZP6 in oocytes using whole-mount in situ hybridization on a fragment of the ovary from *K. alborostrata*. Transcripts of NeZPL1, NeZPL2, and NeZPL4 were identified in the cytoplasm of oocytes (Figure 4A,B,D), whereas transcripts of NeZPL6 were detected in the germinal vesicles of oocytes after an extended reaction time (Figure 4G–J). Transcripts of NeZPL3 and NeZPL5 were not detected in the oocytes (Figure 4C,E), even after extending the reaction time.

### 3.5. Localization of NeZPL6 on the Plasma Membrane of Egg

In light of these results, we focused on NeZPL6 as a candidate protein involved in species-specific interactions between gametes of *K. alborostrata* and *K. fulva* on the egg surface. Accordingly, we investigated whether the signal peptide of NeZPL6 had the ability to localize it to the extracellular region of eggs. We synthesized mRNA containing the 3′ UTR and coding sequence of the signal peptide from NeZPL6 of *K. alborostrata* (KaNeZPL6) followed by the coding sequences of EGFP, the CDS and the 5′ UTR of KaNeZPL6 (Figure 5A). EGFP fluorescence was observed on both the surface and the cytoplasm of mature oocytes injected with mRNA (*n* = 6/6) (Figure 5B). Cytoplasmic fluorescence likely originated from EGFP during exocytosis. This suggests that the signal peptide of NeZPL6 has the ability to transport proteins outside the cell and that endogenous NeZPL6 is localized on the egg surface.

## 4. Discussion

### 4.1. Proteins with Both ZP Modules in Other Lophotrochozoans

In the present study, we identified six genes that encode ZP modules in the transcripts of *K. alborostrata* and *K. fulva*. Of the six genes, NeZPL1, NeZP4, NeZPL5, and NeZPL6 possessed one or two EGF-like domains in addition to the ZP module. A recent study showed that such proteins are commonly found in the genomes and transcripts of several animals, including the nemertean species *Notospermus geniculatus* [20]. These proteins have evolved independently in multiple lineages of lophotrochozoans [20]. In mollusks, such proteins are expressed in the inner part of the mantle epithelium of the pearl oyster, *Pinctada fucata* [20]. A homologous protein is also found in the shells of the nautilus *Nautilus macromphalus* and the limpet *Lottia gigantea* [21,22], indicating that these proteins are related to shell formation in these species. Therefore, the functions of some NeZPL proteins may functionable in other processes than fertilization. Notably, NeZPL3 and NeZPL5 transcripts, which were not detected in oocytes, are likely to be functional in other tissues. In contrast, VERL orthologs are absent not only in *N. geniculatus* but also in other lophotrochozoans, except for mollusks [20], which is consistent with our results that VERL orthologs are absent in transcripts from *K. alborostrata* and *K. fulva*. This indicates that a mechanism different from that observed in mollusks underlies species recognition during fertilization of nemertean species.

### 4.2. Role of NeZPL Proteins on Fertilization

In this study, we identified the transcripts of NeZPL1, NeZPL2, NeZPL4, and NeZPL6 in oocytes. Among these, only NeZPL2 contained repetitive CUB domains other than the ZP module. The CUB domain, initially identified in proteins involved in developmental processes, is also found in sperm adhesin, a protein capable of binding to the ZP in mammals [23,24], and in EBR1, which interacts with sperm in a species-specific manner in sea urchins [25]. The EGF-like domains found in NeZPL1, NeZPL4, and NeZPL6 are common in cell surface proteins and are crucial for cell adhesion [26,27]. In *Caenorhabditis elegans*, the EGF-like domain-containing proteins SPE-9 and SPE-36 are necessary for fertilization in spermatids [28,29]. Therefore, NeZPL proteins expressed in oocytes could likely be involved in the interaction between the gametes of *K. alborostrata* and *K. fulva*. Multiple amino acid substitutions were observed at distant positions in the primary structures between NeZPL proteins of *K. alborostrata* and *K. fulva*.. Similarly, VERL and lysin, which are involved in species-specific interactions in abalones, exhibit polymorphisms at distant positions within their primary structures, although they are closely aligned in their tertiary structures [11]. Furthermore, even a point mutation in the EGF-like domain of Notch alters its ligand-binding capacity [30]. Consequently, the effects of these substitutions in NeZPL proteins on species-specific interactions should be examined in conjunction with their tertiary structures in future studies.

Genes involving species recognition, such as bindin (sea urchin), lysin and VERL (abalone), ADAM10 (coral), and bouncer (teleost), are typically under positive selection [31,32,33,34]. Therefore, although we need to obtain the sequences of NeZPL proteins from other species close to *K. alborostrata* and *K. fulva* for comparison, it will be possible to evaluate the roles of NeZPL proteins from the perspective of molecular evolution.

### 4.3. Effect of the Indelin NeZPL6 on Interaction Between Protein

Here, we found a three-amino acid-indel in the second EGF-like domain of NeZPL6 in *K. alborostrata* and *K. fulva*. Previous studies have shown that indels within proteins affect interactions with other proteins [35]. Moreover, in the ascidians *Halocynthia roretzi* and *H. aurantium*, proteins consisting of repetitive EGF-like domains and a ZP module (HrVC70 and HaVC80) have been identified as candidate molecules involved in self/nonself gamete recognition based on the observed polymorphism among individuals in the EGF-like domains of these proteins [36,37]. HrVC70 is thought to function in conjunction with other proteins containing both EGF-like domains and a ZP module in the VE [38]. Interestingly, some amino acid polymorphisms in HrVC70 and HaVC80 are located in the same region as indel in NeZPL6, between the first and second cysteines of an EGF-like domain [34,35]. The effect of the NeZPL6 indel on nemertean fertilization should be examined in future studies.

## 5. Conclusions

We identified a novel ZP module-containing proteins named NeZPL1 to 6. NeZPL1, 2, 4, and 6 are expressed in oocytes and have amino acid substitutions or an indel between *K. alborostrata* and *K. fulva*, suggesting that they may play roles in species recognition during fertilization. Additionally, we demonstrated the successful expression of exogenous genes in the oocytes of nemertean species. These results contribute to a deeper understanding of fertilization in both nemerteans and lophotrochozoans.

## Figures and Tables

**Figure 1 biomolecules-14-01545-f001:**
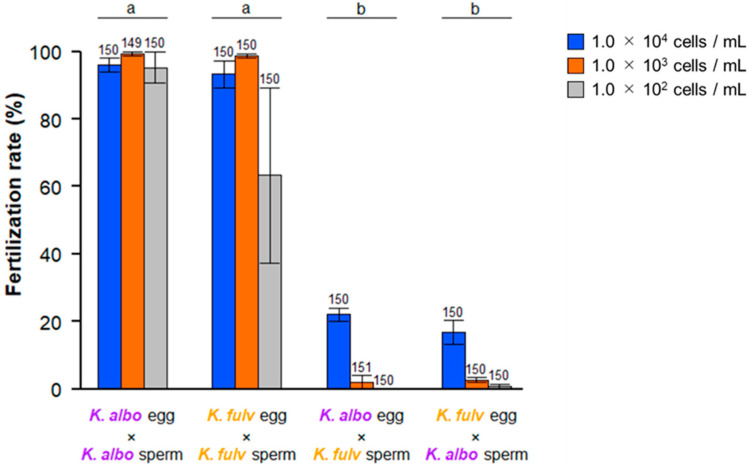
Cross-fertilization rate between gametes from *K. alborostrata* and *K. fulva* under various concentrations of sperm. The number on each graph shows the number of observed eggs. The alphabets on the graphs show the between-group variation by two-way ANOVA (*p* < 0.00001).

**Figure 2 biomolecules-14-01545-f002:**
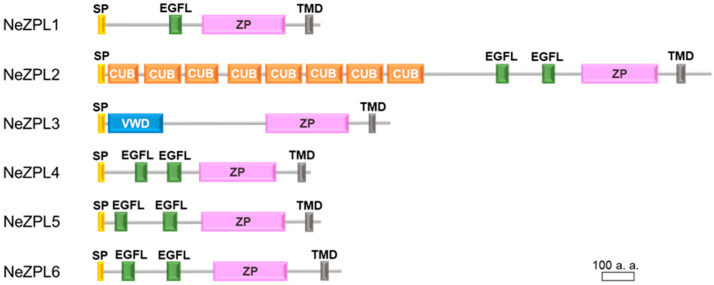
Diagrams of the novel identified proteins NeZPL1–NeZPL6. CUB: CUB domain; EGFL: EGF-like domain; SP: signal peptide; TMD: transmembrane domain; VWD: Von Willebrand factor type D domain; and ZP: ZP module.

**Figure 3 biomolecules-14-01545-f003:**
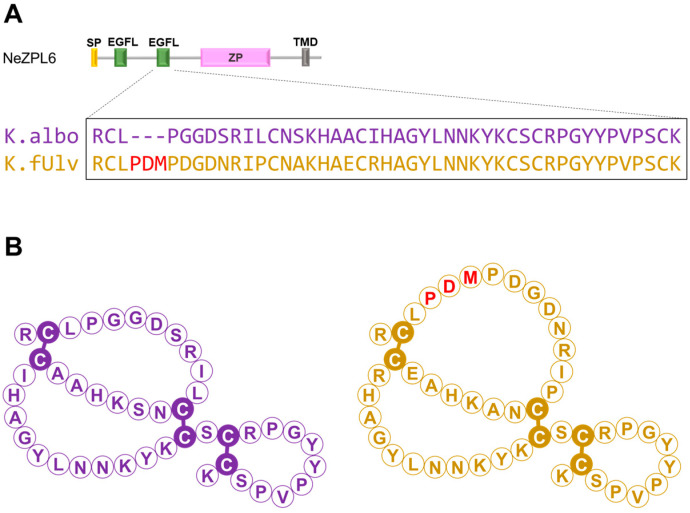
(**A**) Alignment of the amino acid sequence of the second EGF-like domain in *K. alborostrata* (K. albo) and *K. fulva* (K. fulv) NeZPL6. (**B**) Predictive structure of the EGF-like domain of *K. alborostrata* (left) and *K. fulva* (right). The difference in the amino acid sequence is projected onto the general structure of an EGF-like domain.

**Figure 4 biomolecules-14-01545-f004:**
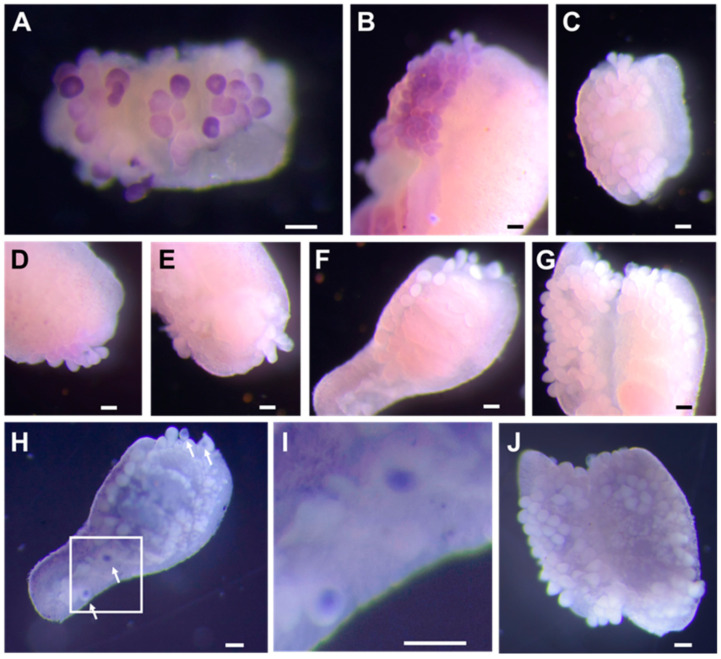
Expression of transcripts of NeZPL1–NeZPL6. (**A**) NeZPL1. (**B**) NeZPL2. (**C**) NeZPL3 (**D**) NeZPL4. (**E**) NeZPL5. (**F**) NeZPL6. (**G**) Common control (sense probe of NeZPL1). (**H**) Staining for NeZPL6 transcript after prolonged reaction. Arrows show signals. (**I**) Magnification of the region enclosed by a white box in (**H**). (**J**) Control for prolonged reaction. Scale bars: 100 µm.

**Figure 5 biomolecules-14-01545-f005:**
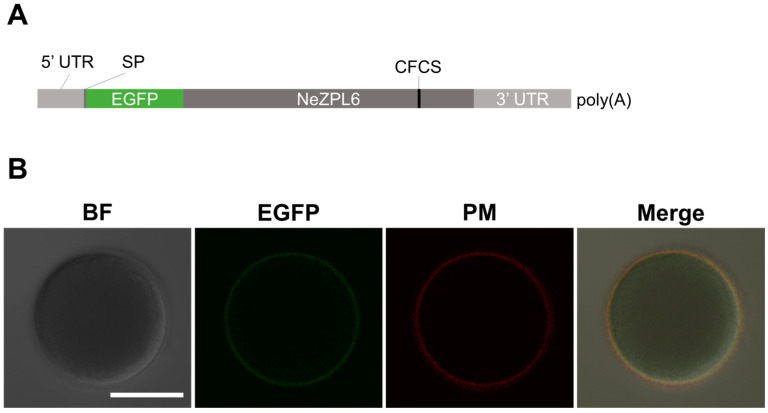
(**A**) Diagram of the synthesized mRNA construct; SP represents the sequence coding a signal peptide. (**B**) Expression of EGFP in oocytes after the mRNA was microinjected. BF indicates Bright Field, and PM indicates the plasma membrane. Scale bar: 50 µm.

**Table 1 biomolecules-14-01545-t001:** The predicted gene coded by the products obtained by cDNA RDA.

Predicted Gene Name	Number of Clones (60 Clones in Total)
vitellogenin	52
abhydrolase	4
cyclin	2
ZP-module containing protein	1
poly-ubiquitin	1

## Data Availability

The transcriptomic data presented in this study are openly available in the Bioproject database (Bioproject ID: PRJNA1180508). The nucleotide sequences of NeZPL1–6 from both *Kulikovia alborostrata* and *K. fulva* are deposited to GenBank with accession numbers PQ653559 343–PQ653570.

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
