# Peer review of "Identification of Six Novel Proteins Containing a ZP Module from Nemertean Species"

_biomolecules, 2024, doi:10.3390/biom14121545_

Round 1
Reviewer 1 Report
Comments and Suggestions for Authors
In this manuscript, the authors identify several proteins containing a ZP module involved in the species-specific gametes interaction of two species of marine nemertean worms. The topic is very interesting since the fertilization process, and in particular, the dynamics of gametes recognition and interaction is still poorly understood. The paper is well-written. However, I have some suggestions that will help to improve the manuscript further.
In particular, I have some concerns regarding the methodology used to assess the occurrence of fertilization. In the Materials and Methods section, the authors stated that the fertilization rate “was calculated 2 h after insemination by counting the eggs during cleavage”. However in my opinion this statement is quite ambiguous and requires clarification. As you know, the fertilization process comprises a series of morphological and biochemical events occurring in both gametes as part of their activation (such as calcium release, sperm penetration in the egg cytoplasm, and fertilization envelope formation, just to name a few). All of them are fundamental to ensure successful embryonic development. Judging the success of fertilization by looking only at embryonic development can be risky because in some cases (especially when the oocyte/egg structure is damaged), oocyte activation can occur without actual or multiple sperm penetration or normal elevation of the fertilization envelope. Even in the absence of polyspermy, the absence of a fertilization envelope around the embryo, (even in an embryo that is dividing symmetrically), does not indicate successful fertilization. Therefore, I suggest the authors include an additional panel, incorporated in Figure 1, showing images of zygotes with the fertilization membrane elevated and the cleavage they evaluated.
Moreover, since in this manuscript, the authors address the cross-fertilization between two species of marine nemertean worms it will be interesting to show the morphological differences existing between gametes (oocyte-mature eggs and spermatozoa) of these two species (if any), to include in the panel I suggested.
Finally, I suggest specifying the number of animals (male and female) from which you collected the oocytes used to perform your experiments. So that the readers can understand if these 150 zygotes for each histogram are from the same or multiple females.
In Figure 5 the confocal images seem to be out of focus and in particular the bright field, which should be supposed to show the oocyte morphology is instead very dark, making it very difficult to distinguish cellular features (such as germinal vesicle), so please substitute with better images. Moreover, please indicate how many oocytes you tested and from how many animals they were collected.
Minor point: reference 18 is missing from the list.
Author Response
General Comments:
In this manuscript, the authors identify several proteins containing a ZP module involved in the species-specific gametes interaction of two species of marine nemertean worms. The topic is very interesting since the fertilization process, and in particular, the dynamics of gametes recognition and interaction is still poorly understood. The paper is well-written. However, I have some suggestions that will help to improve the manuscript further.
Response:
We appreciate your careful reviews and insightful suggestions. We reexamined every point you indicated and made a necessary update following your advices. Point to point answers are below.
Comments1:
In particular, I have some concerns regarding the methodology used to assess the occurrence of fertilization. In the Materials and Methods section, the authors stated that the fertilization rate “was calculated 2 h after insemination by counting the eggs during cleavage”. However in my opinion this statement is quite ambiguous and requires clarification. As you know, the fertilization process comprises a series of morphological and biochemical events occurring in both gametes as part of their activation (such as calcium release, sperm penetration in the egg cytoplasm, and fertilization envelope formation, just to name a few). All of them are fundamental to ensure successful embryonic development. Judging the success of fertilization by looking only at embryonic development can be risky because in some cases (especially when the oocyte/egg structure is damaged), oocyte activation can occur without actual or multiple sperm penetration or normal elevation of the fertilization envelope. Even in the absence of polyspermy, the absence of a fertilization envelope around the embryo, (even in an embryo that is dividing symmetrically), does not indicate successful fertilization. Therefore, I suggest the authors include an additional panel, incorporated in Figure 1, showing images of zygotes with the fertilization membrane elevated and the cleavage they evaluated.
Response1:
Thank you for your suggestion. As you indicated, we also think it is important to evaluate with multiple criteria whether embryogenesis occurs normally. However, in this case of Kulikovia alborostrata and K. fulva, we would like to note that the fertilization membrane is not recognized after fertilization even under high magnificence. This seems to be normal for these species because we observed those embryos develop normally to swimming larva. Therefore, we used the cleavage rate as criterion for fertilization. In our experience, eggs of these species do not show spontaneous cleavages without meeting sperm. Meanwhile, when there is excess sperm around eggs, some eggs show abnormal cleavage possibly because of polyspermy. In fact, in our data of Fig. 1, a few eggs under 1.0 x 104 sperm/mL showed abnormal cleavage, but were counted as fertilized with a thought that these succeeded in species-specific recognition and fertilized. For these reasons, we unfortunately cannot show images of fertilization membrane and general cleavage as you suggested, we think the presence or cleavage of eggs is sufficient for evaluation concerning the two species.
Comments 2:
Moreover, since in this manuscript, the authors address the cross-fertilization between two species of marine nemertean worms it will be interesting to show the morphological differences existing between gametes (oocyte-mature eggs and spermatozoa) of these two species (if any), to include in the panel I suggested.
Response 2:
Thank you for important advice. We were also interested in this point and reported in our previous research (Ikenaga et al., 2021; see Fig. 1). As a result, there was almost no morphological difference between gametes of the two species. We added this information to Introduction section. Please see lines 54 to 56.
Comments 3:
Finally, I suggest specifying the number of animals (male and female) from which you collected the oocytes used to perform your experiments. So that the readers can understand if these 150 zygotes for each histogram are from the same or multiple females.
Response 3:
Thank you very much for this suggestion. This result bases on a single female and single male. We added this information to the Materials and Methos section. Please see line 66.
Comments 4:
In Figure 5 the confocal images seem to be out of focus and in particular the bright field, which should be supposed to show the oocyte morphology is instead very dark, making it very difficult to distinguish cellular features (such as germinal vesicle), so please substitute with better images. Moreover, please indicate how many oocytes you tested and from how many animals they were collected.
Response4:
We appreciate this important remark. We replaced these images to another ones. We would like to note that we cannot detect GV in these images because these are images of oocytes after GVBD. We also this information to the Materials and Method section. Please see lines 170 and 172. As to numbers of individuals and oocytes, we observed six oocytes from a single female. We also added this information to line 169, 170 and 252.
Comments 5: Minor point: reference 18 is missing from the list.
Response 5: Thank you very much for you careful reviewing. We fixed it.
Reviewer 2 Report
Comments and Suggestions for Authors
This study reports the identification of 6 ZP domain containing proteins in 2 nemertean species, their mRNA expression in oocytes, and the localization of one the proteins, NEZPL6, on the oocyte surfaces. The manuscript is fairly well written, and the results are well presented. The authors may consider the following comments.
1. Any justification for using 2 nemertean species in this study?
2. Line 75, Here “fragment” contains oocytes or spermatozoa?
3. The method for cDNA representational difference analysis needs some details. It is not clear how this was done. Table 1 title indicates ovary and testis were used in cDNA RDA but this information is missing in the cDNA RDA method.
4. Line 175-176, The information of these full sequences is missing. RACE is not clear. It includes both 5RACE and 3RACE?
5. LINE 178 “assembled transcripts from the ovaries of K. alborostrata and K. fulva”. Was cDNA RDA only performed on K. alborostrata?
6. The information about the cDNA sequences and the predicted amino acids sequences of the 6 proteins is not provided. If they are novel sequences, they should be deposited in the Genbank.
7. The study used the sequence of the ZP module of genes identified by cDNA RDA as a query to search the database. Is the ZP module sequence known? If this is known, can this be used directly in blast search without the need for cDNA RDA?
8. Can the expression of mRNA for 6 genes in oocytes be analyzed by RT-PCR?
9. Figure 5A, should 5UTR be upstream of SP and EGFP?
Author Response
General Comments:
This study reports the identification of 6 ZP domain containing proteins in 2 nemertean species, their mRNA expression in oocytes, and the localization of one the proteins, NEZPL6, on the oocyte surfaces. The manuscript is fairly well written, and the results are well presented. The authors may consider the following comments.
Response:
We appreciate your reviewing on our manuscript and valuable indications. Our point-to-point answers to your suggestions are below.
Comments 1.
Any justification for using 2 nemertean species in this study?
Response 1:
Yes. We think following things will justify the use these species for the study; (i) there is nothing known about species-recognition in species of Lophotrochozoa sensu stricto and (ii) these are reproductively separated even though genetically close to each other. Please see lines 50 to 54.
Comments 2.
Line 75, Here “fragment” contains oocytes or spermatozoa?
Response 2:
We are grateful to your remark. Yes, this indicates a fragment containing oocytes. We revised the sentence. Please see line 78.
Comments 3.
The method for cDNA representational difference analysis needs some details. It is not clear how this was done. Table 1 title indicates ovary and testis were used in cDNA RDA but this information is missing in the cDNA RDA method.
Response 3:
We appreciate your advice and must say sorry our mistakes in our materials and methods. After we referred again our records of experiments, we found we conducted cDNA RDA with cDNAs from mature and immature ovary. We updated the Materials and Methods section and corrected related part. Please see lines 84 to 86 and 215.
Comments 4.
Line 175-176, The information of these full sequences is missing. RACE is not clear. It includes both 5RACE and 3RACE?
Response 4:
Thank you for the remarks. Yes, we conducted both 5’ and 3’ RACE. We described this point in detail. Please see lines 100 to 102.
Comments 5.
LINE 178 “assembled transcripts from the ovaries of K. alborostrata and K. fulva”. Was cDNA RDA only performed on K. alborostrata?
Response 5:
Yes, cDNA RDA was conducted only on K. alborostrata, but de novo transcript analysis was conducted both on K. alborostrata and K. fulva. Concerning your indication No.7, we revised description related to the cDNA RDA and BLAST. Please see lines 192 to 200.
Comments 6.
The information about the cDNA sequences and the predicted amino acids sequences of the 6 proteins is not provided. If they are novel sequences, they should be deposited in the Genbank.
Response 6:
Thank you for pointing out and we are sorry for missing this information. We added the Genbank ID for these genes in the materials and method section. Please see line 342 and 343. Because we have not received the numbers yet, we replace true numbers with “XXXX” for now. We will update this as soon as we receive them.
Comments 7.
The study used the sequence of the ZP module of genes identified by cDNA RDA as a query to search the database. Is the ZP module sequence known? If this is known, can this be used directly in blast search without the need for cDNA RDA?
Response 7:
Thank you for remark for sophisticating our manuscript. The sequence of ZP module is known in other species even though homology to that of these species is not so high, but as you mentioned, these genes will be derived in the blast with the other species’ ZP-module sequence as query. We would note that we conducted that the cDNA RDA to detect highly expressed genes in ovary and the BLAST to explore other ZP-module containing genes. This was encouraged by the knowledge in vertebrates that one ZP-module containing proteins are usually conjugated with other ZP-module containing proteins. We revised the sentences relates and made clear these aims. Please see lines 192 to 200.
Comments 8.
Can the expression of mRNA for 6 genes in oocytes be analyzed by RT-PCR?
Response 8:
We are grateful to this indication. Unfortunately, we tried this but couldn’t purify mRNA from oocytes possibly because of jelly surrounding eggs. In our thought, mRNA of these genes may not be able to be detected in mature oocytes because formation of ZP will complete before oocytes’ maturation.
Comments 9.
Figure 5A, should 5UTR be upstream of SP and EGFP?
Response 9:
Thank you very much for pointing this out. This is our mistake. We apologize for shortage of our revision. We have corrected it.